# Combined Effect of Caspase-Dependent and Caspase-Independent Apoptosis in the Anticancer Activity of Gold Complexes with Phosphine and Benzimidazole Derivatives

**DOI:** 10.3390/ph14010010

**Published:** 2020-12-24

**Authors:** Lara Rouco, Ángeles Sánchez-González, Rebeca Alvariño, Amparo Alfonso, Ezequiel M. Vázquez-López, Emilia García-Martínez, Marcelino Maneiro

**Affiliations:** 1Departamento de Química Inorgánica, Facultade de Ciencias, Universidade de Santiago de Compostela, 27002 Lugo, Spain; lara.rouco.mendez@usc.es; 2Departamento de Química Inorgánica, Facultade de Farmacia, Universidade de Santiago de Compostela, 15782 Santiago de Compostela, Spain; 3Departamento de Farmacología, Facultade de Veterinaria, Campus Terra, Universidade de Santiago de Compostela, 27002 Lugo, Spain; amparo.alfonso@usc.es; 4Departamento de Química Inorgánica, Facultade de Química, Campus Universitario Lagoas-Marcosende, Universidade de Vigo, 36310 Vigo, Spain; ezequiel@uvigo.es (E.M.V.-L.); emgarcia@uvigo.es (E.G.-M.)

**Keywords:** gold(I) compounds, cytotoxic activity, caspase, apoptosis, neuroblastoma SH-SY5Y

## Abstract

Since the potential anticancer activity of auranofin was discovered, gold compounds have attracted interest with a view to developing anticancer agents that follow cytotoxic mechanisms other than cisplatin. Two benzimidazole gold(I) derivatives containing triphenylphosphine (Au(pben)(PPh_3_)) (**1**) or triethylphosphine (Au(pben)(PEt_3_)) (**2**) were prepared and characterized by standard techniques. X-ray crystal structures for **1** and **2** were solved. The cytotoxicity of **1** and **2** was tested in human neuroblastoma SH-SY5Y cells. Cells were incubated with compounds for 24 h with concentrations ranging from 10 µM to 1 nM, and the half-maximal inhibitory concentration (IC_50_) was determined. **1** and **2** showed an IC_50_ of 2.7 and 1.6 µM, respectively. In order to better understand the type of cell death induced by compounds, neuroblastoma cells were stained with Annexin-FITC and propidium iodide. The fluorescence analysis revealed that compounds were inducing apoptosis; however, pre-treatment with the caspase inhibitor Z-VAD did not reduce cell death. Analysis of compound effects on caspase-3 activity and reactive oxygen species (ROS) production in SH-SY5Y cells revealed an antiproliferative ability mediated through oxidative stress and both caspase-dependent and caspase-independent mechanisms.

## 1. Introduction

The pharmacological use of gold compounds for the treatment of different diseases has been documented since ancient times [1,2]. More recently, in the twentieth century, the antiarthritic properties for Au(I) thiolates were observed [2], but their adverse effects led to the development of gold complexes with phosphine ligands, which are more lipophilic and remain in circulation for a longer time [3]. Auranofin, 2,3,4,6-tetra-*O*-acetyl-l-thio-β-d-glyco-pyranosato-*S*-(triethylphosphine) gold(I) is an example of these Au(I) thiolates-phosphine derivatives (see Scheme 1A), which contain a S-Au-P fragment, used as disease-modifying anti-rheumatic drugs [3,4,5] but also investigated for other potential therapeutic applications, from neurodegenerative disorders [6] or bacterial infections [7] to the inhibition of the novel SARS-COV-2 coronavirus [8].

The anticancer activity of this type of compound has also attracted interest due to its cytotoxic activity against cells from several tumor cell lines [9,10,11]. Among them, ovarian carcinoma [12], colorectal cancer [13], cervical epithelial carcinoma [14], and lung cancer cells [15]. The mechanism associated with this therapeutic activity has not yet been totally established, but results point to the ability of the Au(I) complexes to interact with biological targets such as proteins or nucleic acids [16,17] and, particularly, with thiol-containing proteins like the thioredoxin reductase (TrxR) [18,19], an enzyme involved both in the defense against oxidative damage and in the redox signaling, being relevant for cell growth and development [20]. Other studies have clearly shown the proteasome as a primary target for these complexes [21] or the inhibition of other enzymes such as glutathione reductase or peroxidase [22]. The anticancer activity of these gold compounds seems to follow different mechanisms than cisplatin or other Pt(II)/Pd(II) compounds, which mainly act by binding to DNA and inhibiting its replication. Gold compounds also exhibit cytotoxic activity against cisplatin-resistant cell lines [9,10].

The increasing interest in developing different types of S-Au-P gold-based anticancer candidates led to new synthetic drugs rather than the former pyranosidic derivatives. In this sense, gold complexes incorporating ligands such as thiosemicarbazone [14,23,24], thiolate [25,26], dithiocarbamate [27], and sulfanylcarboxylates [28] have been probed to inhibit thioredoxin reductase by binding the selenocysteines or the cysteinyl thiols of the enzyme.

The replacement of the S-donor ligands by N-donor ligands produced a new family gold complexes that have revealed interesting anticancer activities. These N-donor ligands include imidazoles [29], naphthalimides [30], cytosines [31], bipyridines [32], phenanthrolines [33], or benzimidazoles [34,35,36]. The biochemical mechanism of action involved in the anticancer activity of the gold complexes with N-donor ligands is presumably the same as that of auranofin, although it is a matter of study interest due to the diversity of results obtained in different investigations regarding biological targets for auranofin [18,19,20,21,22], and due to the different complex stabilization by the replacement of the thiolate ligand by the N-donor ligand.

In the present study, we propose to use 2-(2′-pyridyl)benzimidazole (Hpben) as a ligand with potentially three N-donor atoms (see Scheme 1B), which would lead to different coordinations with the gold atom [35]. Benzimidazoles are very versatile ligands that show various pharmacological properties such as antibacterial, antiviral, and anti-inflammatory [37,38]. Our aim is to obtain N-Au-P complexes, for which Hpben and two different phosphines are used: triphenylphosphine and triethylphosphine. Human neuroblastoma SH-SY5Y is the cell line chosen to study the cytotoxicity, the type of cell death in the anticancer activity, and the mechanism of antiproliferative action of these gold complexes.

## 2. Results

### 2.1. Analytical and Spectroscopic Characterization of ***1*** and ***2***

Complexes **1** and **2** were obtained in high yield as detailed in the experimental section (Scheme 2). Their structures were fully characterized using ^1^H, ^13^C and ^31^P NMR spectroscopy, IR and UV spectroscopy, electrospray ionization mass spectroscopy (ESI-MS), and single-crystal X-ray crystallography. Elemental analyses establish a formula (Au(pben)(PR_3_)), being R=Ph for **1** and R=Et for **2**. ESI-MS spectra confirmed these formulas for the neutral complexes by showing molecular ion peaks at *m*/*z* 644.14 for **1** and m/z 510.13 for **2** (Appendix A). UV-Vis spectra showed in DMSO a common absorption band at 313 nm assigned to π–π* transitions located in the heteroaromatic benzimidazole rings (Appendix A). Complex **1** displays an additional band at 325 nm, which has also previously been found in gold(I) complexes containing the ancillary PPh_3_ ligand [34].

The IR spectra of these complexes (Appendix A), when compared with the corresponding spectrum of its free ligand, show the absence of the ν(N–H) band, which is present in the spectrum of the Hbpen between 2990–2700 cm^−1^, consistent with the deprotonation of this group and the N-coordination to the Au atom. In both complexes, the aromatic tension bands of the pben ligand do not undergo significant changes when coordination occurs, although rigorous assignment is made difficult by the great intensity showed by the bands from the phosphines.

The ^1^H NMR spectra of **1** and **2** showed each one as well-resolved set of signals (Appendix A); the H6′ proton at δ 8.57/8.52 was the most deshielded and downfield shifted of 0.13–0.18 ppm with respect to the free ligand (^1^H NMR and ^13^C NMR spectra for free ligand are collected in Appendix A), while the resonance of the H3′ proton was shifted at about 0.18 ppm, which suggests some involvement of the pyridinic nitrogen in the Au-pben bonding. In both complexes, the N–H signal of the free ligand disappeared, confirming the deprotonation of Hbpen. In the spectrum of **1**, the signals corresponding to the co-ligand PPh_3_ overlap with those of pben^−^, so the assignment is tentative. Aromatic carbons from pben^−^ and triphenylphosphine ligands are registered in the ^13^C NMR spectrum of **1** (Appendix A). ^31^P NMR spectrum of **1** shows a single signal at 30.9 ppm, which is evidence that the gold complex is not altered in solution (Appendix A). In the case of **2**, the identification and assignment of PEt_3_ signals were unambiguous with the –CH_3_ signals as a doublet of triplets at 1.31 ppm due to the coupling with ^31^P (coupling constant ^3^*J*(H–P) = 18.5 Hz, slightly higher than the corresponding to the free PEt_3_, 14 Hz). The ^13^C NMR spectrum of **2** showed two signals (Appendix A) corresponding to the phosphine at 9.2 (CH_3_) and 18.5 (d, CH_2_, *J*(C–P) = 37.3 Hz) ppm, δ and J values which are like those found for other Au(I)-PEt_3_ complexes [14,28], and that contrast with the values of non-coordinated PEt_3_. ^31^P NMR spectrum of **2** showed a single signal at 26.9 ppm (Appendix A).

The electrochemical behavior of compounds **1**, **2** was investigated in DMSO-tetraethylammonium perchlorate 0.1 M solvent system through cyclic voltammetry. Figure 1 shows the voltammograms at different scan rates. The Au(I) gold complexes undergo one oxidation process (at −0.594 for **1** and at −0.597 V for **2**) to form the Au(III) species.

The redox properties of the two gold(I) complexes showed grossly similar behavior consisting of a quasi-reversible Au(I)/Au(III) oxidation. The criteria of reversibility were checked by observing the constancy of peak-peak separation (ΔE_p_ = E_pa_ – E_pc_) of 188 mV for **1** and 214 mV for **2** at 0.02 V s^−1^. The reversible character decreased as scan rates increased. Thus, ΔE_p_ are 281 mV for **1** and 259 mV for **2** at 0.05 V s^−1^, and even 344 mV for **1** and 297 mV for **2** at 0.09 V s^−1^. As is predictable, the intensity of the current also increased at higher scan rates. Cyclic voltammograms of **1**,**2** endorse the purity and stability of both complexes in solution.

### 2.2. Crystal Structures of Complexes ***1*** and ***2***

Single crystals of complex **1** suitable for X-ray diffraction studies were obtained by slow evaporation of the methanolic solution at room temperature. The molecular structure is shown in Figure 2 and Appendix A, and the main bond distances and angles are shown in Table 1.

The asymmetric unit contained two molecules of **1**, (Au(pben)PPh_3_), with slightly different bond distances and angles values and one disordered methanol molecule. Since it did not establish any relevant interaction with (Au(pben)PPh_3_), it was removed using the SQUEEZE program [39]. In the two molecules of the complex, the metal ion is coordinated to the phosphorous atom of triphenylphosphine and to the nitrogen atom of the benzimidazolate ring with an almost linear disposition. An additional contact was established between the nitrogen atom of the pyridine and the gold ion, giving rise to a 2 + 1 coordination number.

The Au-P Bond lengths (2.230(2) y 2.235(2) Å) were similar to those found in the cationic complex (Au(Hpben)(PPh_3_))ClO_4_ [40]. However, the Au-N bond distances (2.058(6) and 2.060(6)) were slightly shorter than those found in the (Au(Hpben)(PPh_3_))^+^ cation (2.075(4) Å), consistent with a stronger interaction between the metal ion and the deprotonated ligand. The bond length between the gold and the pyridine nitrogen (Au(1)···N(3) = 2.729(7) Å and Au(2)···N(6) = 2.782(7) Å) was shorter than the sum of the Van der Waals radii (3.25 Å), and shorter than that observed in (Au(Hpben)(PPh_3_))ClO_4_ (2.930 Å) [40]. Bond lengths and angles found in the triphenylphosphine ligands (not collected in Table 2) showed normal values and similar to those reported for complexes of Au(I) and PPh_3_ [40,41,42].

Single crystals of **2** were obtained as detailed in the experimental section; a perspective view is shown in Figure 3 and Appendix A, and selected bond parameters are collected in Table 2. The asymmetric unit of the crystal structure of **2** also consists of two chemically equivalent molecules in which the benzimidazolate unit is acting as a monodentate ligand through the N(1) nitrogen atom. The gold(I) ion was also arranged in the usual almost linear coordination geometry, coordinated to pben and to the phosphorous atom of triethylphosphine. In the same way as in **1**, an additional contact was established between the nitrogen atom of the pyridine and the gold ion, giving rise to a 2 + 1 coordination number. Au-P Bond lengths (2.2288(18) and 2.224(2) Å) and Au-N bond distances (2.050(6) and 2.058(5)) are values very close to those found in **1**.

### 2.3. Cytotoxicity Studies and Determination of Cell Death Type

SH-SY5Y neuroblastoma cells were used to study the toxic effect of the Au(I) complexes. Cells were treated with different concentrations of **1** and **2** at two incubation times, 6 and 24 h. Table 3 shows IC_50_ values calculated for compounds (see also Appendix A). Complex **2** was more cytotoxic, presenting lower IC_50_ values than complex **1** at both times.

Next, to determine the type of cell death produced by gold complexes, their effects on neuroblastoma cells were evaluated using the fluorescent dyes Annexin V-FITC (fluorescein isothiocyanate) and propidium iodide (PI), which allow the detection of necrotic and apoptotic cell populations by flow cytometry. Three populations were discriminated: viable cells (Annexin V-FITC and PI negative cells), apoptotic cells (including Annexin V-FITC positive and PI negative cells, and Annexin V-FITC and PI-positive cells), and necrotic cells (Annexin V-FITC negative and PI-positive cells). In this assay, SH-SY5Y cells were treated with compounds at IC_50_ concentrations for 6 and 24 h. Furthermore, to check if the complexes **1** and **2** were inducing caspase-dependent apoptosis, cells were pre-incubated with the pan-caspase inhibitor Z-VAD-FMK (carbobenzoxy-valyl-alanyl-aspartyl-[*O*-methyl]-fluoromethylketone) at 40 μM for 24 h, followed by treatment with compounds at IC_50_ concentrations for 6 and 24 h. In order to validate the model, staurosporine (STS) was used as cell death control (Figure 4).

Treatment with compounds **1** and **2** for 6 h produced a significant decrease in viable cells, with levels of 70.6 ± 3.0% and 74.1 ± 1.4%, respectively (*p* < 0.05) (Figure 4A). In this assay, gold compounds produced a lower decrease in cell viability than expected, which could be due to the differences between the 3-(4,5-dimethyl thiazol-2-yl)-2,5-diphenyl tetrazolium bromide (MTT) assay and Annexin/PI staining. IC_50_ values were determined with MTT, which quantifies cell viability based on mitochondrial activity, so it was more sensitive than the Annexin/PI staining, which detects changes in the plasmatic membrane [43,44,45]. Both complexes produced an increase in apoptotic cells (15.0 ± 1.2% for **1** and 13.7 ± 1.3% for **2**). Moreover, compound **1** significantly augmented necrotic cells (14.4 ± 1.0%) compared to control cells. As expected, the addition of 0.1 µM STS, a known apoptotic inducer, generated an increase of apoptotic cells (29.3 ± 3.1%).

On the other hand, cells co-treated with Z-VAD and compounds **1** and **2** for 6 h presented a significant decrease in viable cells (71.6 ± 2.7% and 74.3 ± 2.1%, respectively). Interestingly, the percentages of apoptotic cells after pre-treatment with Z-VAD (14.1 ± 0.7% and 13.0 ± 0.9%, respectively) were similar to those obtained with the compounds alone. Otherwise, the apoptotic death produced by 0.1 µM STS alone was significantly reduced when neuroblastoma cells were pre-treated with the caspase inhibitor (18.5 ± 1.1%, *p* < 0.05). Regarding necrosis, its levels were comparable to those observed with gold complexes alone, with percentages among 12.7–14.3%. In the case of STS, necrotic cells were significantly reduced by pre-treatment with Z-VAD. The results obtained with STS validated the model since its cytotoxic effects were reduced when the caspase inhibitor was added to SH-SY5Y cells. However, Z-VAD was unable to reduce the cell death produced by gold compounds.

This assay was repeated after a longer incubation time, 24 h (Figure 4B). A reduction in cell survival was observed when cells were treated with complexes **1** and **2** (68.1 ± 0.9% and 59.3 ± 2.0%, respectively). Once again, compounds produced a significant augmentation of apoptotic cells, with levels of 22.8 ± 0.6% after the addition of complex **1**, and 27.0 ± 0.5% when cells were treated with **2**. In order to further confirm the results obtained at 6 h, neuroblastoma cells were pre-treated with 40 µM Z-VAD, followed by the addition of complexes **1** and **2** for 24 h. In this assay, both compounds produced a significant decrease in viable cells (67.5 ± 2.1% and 66.2 ± 3.3%, respectively). In addition, the levels of apoptotic cells were between 19.9–23.2%, confirming the inability of Z-VAD to inhibit the death of neuroblastoma cells. With respect to STS, pre-treatment with Z-VAD did not produce a significant reduction in the apoptotic death generated by the compound after 24 h. This effect has been previously described and is related to the ability of STS to produce caspase-dependent and caspase-independent apoptosis. The first one happens at shorter incubation times, whilst the second one occurs more slowly and can be observed at longer incubation times [46,47]. In our model, Z-VAD reduced the apoptosis produced by STS after 6 h of incubation, but no effect was found at 24 h, agreeing with these previous results.

In the case of gold complexes, previous studies have reported that this class of compounds can also induce apoptosis through both caspase-dependent and caspase-independent mechanisms [48,49]. In this context, the effect of complexes **1** and **2** on caspase-3 activity was analyzed. SH-SY5Y cells were treated with compounds at IC_50_ concentrations for 6 and 24 h, and the enzymatic activity was determined (Figure 5).

At 6 h, STS produced an augmentation in caspase-3 activity of 242.5 ± 7.2% (*p* < 0.001), agreeing with its previously reported effect on caspase-dependent apoptosis at short incubation times [46,47]. With regard to gold complexes, only **2** showed a significant increase in enzyme activity (190 ± 10.3%; *p* < 0.001). The assay was repeated for 24 h and both complexes augmented caspase-3 activity, 129.0 ± 10.5% (*p* < 0.05) and 140.8 ± 12.3% (*p* < 0.01), respectively. At this time, STS also produced a significant increase in caspase-3 activity (170.5 ± 6.8%, *p* < 0.001). This rise was smaller than the observed at 6 h, confirming that the involvement of caspase-3 in STS-induced death is lesser at 24 h. With regard to gold complexes, compound **2** presented a similar behavior to STS, with a lower increase in caspase-3 activity at 24 h, whereas complex **1** produced a slight augmentation of caspase-3 activity. These results could explain the lack of effect of pre-treatment with Z-VAD in our previous assays, pointing to a combined effect of the compound on caspase-dependent and caspase-independent apoptosis.

Previously published studies [48,49] stated that Au(I) and Au(III) complexes produce cell death through the induction of oxidative stress. Therefore, the effect of compounds on ROS production was analyzed. Cells were co-treated with **1** and **2** at nine concentrations (0.01–10 µM) for 24 h, and the levels of these damaging molecules were evaluated with a fluorometric assay (Figure 6). Treatment with complex **1** significantly augmented ROS production, reaching levels of 122.4 ± 11.9% (*p* < 0.05) at 5 µM compared to control cells. Compound **2** also produced a significant increase in ROS release at three concentrations (0.01, 0.5, and 7.5 µM), being the greater augmentation at 7.5 µM (129.7 ± 13.8%, *p* < 0.05). The increase generated by gold complexes was comparable to the rise produced by the known pro-oxidant tert-butyl hydroperoxide (TBHP) at 75 µM (134.9 ± 7.1%, *p* < 0.001).

## 3. Discussion

The set of techniques used to characterize **1** and **2** both in solid-state and in solution has made it possible to establish the high purity of both compounds and their stability in solution. Structural characterization did not allow the differentiation of any property between compounds that caused a difference in cytotoxicity. However, complex **2**, which incorporates triethylphosphine co-ligand, shows more cytotoxicity than **1**, with a triphenylphosphine co-ligand. The same influence of substitution of PPh_3_ for PEt_3_ on cytotoxicity has been already reported before for mono phosphinegold(I) sulfanylcarboxylates [28].

Different electrochemical properties have been reported for other anticancer gold complexes [34,35,36], but, in the present case, the electrochemical behavior of **1** and **2** is basically similar, according to both their redox potential values and their quasi-reversible character, so we can conclude that this should not be the factor that determines the different toxicity of both compounds. The greatest steric effect of the triphenyl substituents on phosphine in **1** with respect to the triethyl substituents in **2** arises as a possible explanatory factor that would explain the different antiproliferative action of these gold complexes.

One of the main interest in anticancer gold complexes is the development of pharmacological candidates with cytotoxic activity against cells from tumor lines which are resistant to the chemotherapeutic drug cisplatin or new drugs that may induce apoptosis in cancer cells through different mechanisms than cisplatin or other Pt(II)/Pd(II) compounds [9,10]. The cytotoxicity of cisplatin and oxaliplatin have been previously reported on the human neuroblastoma cell line SH-SY5Y [50]. IC_50_ values for these platinum compounds of 15–50 μM using the MTT assay at an incubation time of 24 h are significantly higher than values obtained for gold(I) complexes **1** and **2** (IC50 of 1.6–2.7 μM), which behaved as more cytotoxic.

In the case of the cisplatin derivatives, their cytotoxic activity has been linked to their ability to crosslink with the purine bases on the DNA. In the present study, the results obtained in the caspase-3 assay and in the determination of ROS production indicate that gold complexes could have antitumor behavior, being able to induce apoptotic cell death through the increase of ROS levels. The redox potentials of **1** and **2** are outside the biologically accessible redox potential window of −0.4 to +0.8 V, suggesting that these complexes do not directly generate ROS due to the Au(I)/Au(III) process but rather as a consequence of inhibition of TrxR [48]. This inhibition damages the thioredoxin system, an important cellular antioxidant defense, increasing ROS levels, and leading to cell death [51,52,53]. The apoptotic cell death produced by complexes **1** and **2** seems to be mediated by two mechanisms, both caspase-dependent and caspase-independent. This dual behavior has been reported for other gold complexes [49] and could be related to the increase in ROS levels produced by compounds. Elevated ROS causes mitochondrial membrane depolarization, which produces a disruption of the electron transport chain, leading to the release of cytochrome c and the apoptosis-inducing factor (AIF) from the mitochondria. Cytochrome c release induces caspase-dependent apoptosis, mediated by caspase-3 and caspase-9 activation, whilst AIF is translocated to the nucleus and produces caspase-independent apoptosis [48].

## 4. Materials and Methods

### 4.1. Materials

Triphenylphosphinegold(I) chloride (99.9%, Strem Chemicals), Triethylphosphinegold(I) chloride (99%, ABCR) and 2-(2′-pyridyl)benzimidazole (Hpben; 99.8%, Aldrich, Darmstadt, Germany) were used as supplied.

### 4.2. Physical Measurements

Elemental analyses of C, H, and N were performed on a Carlo Erba EA-1108 (CE Instruments, Wigan, UK). IR spectra were recorded as KBr discs on a Bio-Rad FTS135 spectrophotometer (Bio-Rad Laboratories, Hercules, CA, USA) in the range 4000–400 cm^–1^. IR data are reported in the experimental section following abbreviations: vs = very strong; s = strong; m = medium; w = weak; sh = shoulder; br = broad.

^1^H NMR and ^13^C NMR spectra were recorded on a Bruker AC-300 spectrometer (Bruker BioSpin, Rheinstetten, Germany) at 296 K using DMF-d_7_ or CDCl_3_ as deuterated solvents. ^1^H NMR and ^13^C NMR were recorded in δ units relative to deuterated solvent as an internal reference. ^31^P NMR spectra were recorded in CDCl_3_ at 202.46 MHZ on a Bruker AMX 500 spectrometer (Bruker Analytik, Karlsruhe, Germany) using 5 mm o.d. tubes and are reported to external H_3_PO_4_ (85%). The following abbreviations were used as s = singlet; d = doublet; t = triplet; m = multiplet.

Positive electrospray ionization (ESI) mass spectra of the ligands and complexes were recorded on a LC-MSD 1100 Hewlett-Packard instrument (Hewlett-Packard, Palo Alto, CA, USA) (positive-ion mode, 98:2 CH_3_OH/HCOOH as the mobile phase, 30–100 V). Electronic spectra were recorded on a Cary50 spectrometer.

Electrochemical measurements were performed using an Autolab PGSTAT101 (Metrohm Autolab, Kanaalweg, The Netherlands) using a three-electrode configuration. The working electrode was a Metrohm model 6.1204.300 graphite disc, while a Pt wire and an Ag/AgCl electrode served as counter and reference electrodes, respectively. Measurements were made with ca. 10^−3^ M solutions of complexes in DMF using 0.1 M tetraethylammonium perchlorate as a supporting electrolyte.

### 4.3. Synthesis of the Complexes

Complexes were prepared through ligand deprotonation with sodium hydroxide, previous to the addition of the gold salt in stoichiometric relation [28,35]. Experimental procedure and characterization data for **1** and **2** are collected below.

#### 4.3.1. (Au(pben)(PPh_3_)) (**1**)

To a stirred solution of (AuCl(PPh_3_)) (200 mg, 0.4 mmol) in methanol (10 mL) was added a methanolic solution (4 mL) of 2-(2′-pyridyl)benzimidazole (Hpben, 79 mg, 0.4 mmol) and NaOH (16 mg, 0.4 mmol). The yellow solution formed was stirred in absence of light for 24 h at room temperature. The crystalline white solid formed was filtered, washed with cold methanol, and dried under vacuum. Yield: 83%. M.P.: 158–160 °C. Anal Calcd. (%) for C_30_H_23_N_3_PAu (653.46 g mol^−1^): C, 55.1; N, 6.4; H, 3.6. Found: C, 55.0; N, 6.4; H, 3.7. MS ES (*m*/*z*) 654.14 (Au(pben)(PPh_3_))^+^. IR (KBr, cm^−1^): ν(C–H) 3050w; ν(PPh_3_) 1590 m, 1500 m, 1420 s; 1230 m, 1160 m, 1045 m, 820 m. UV-vis (DMSO): λ_max_: 313, 325 nm. ^1^H NMR (CDCl_3_, ppm): δ 8.57 (d, 1H, H6′), 8.20 (d, 1H, H3′), 7.5–7.9 (m + m + m, 4H, H5′-H7-H4-H4′), 7.5–7.7 (m + m + m, 15H, PPh_3_), 7.20 (m, 2H, H5-H6). ^31^P NMR (CDCl_3_, ppm) 30.93 (s). E_ox_ (at 0.02 V s^−^^1^) = −0.594 V; E_red_ (at 0.02 V s^−^^1^) = −0.782 V.

#### 4.3.2. (Au(pben)(PEt_3_)) (**2**)

A solution of Hpben (83.5 mg, 0.428 mmol) and NaOH (17.1 mg, 0.428 mmol) in methanol (4.3 mL) was added to a stirred solution of (AuCl(PEt_3_)) (150 mg, 0.428 mmol) in methanol (3 mL). The resulting mixture was stirred in the dark for 48 h at room temperature. Afterward, the solution was cooled to 4 °C, the solvent was evaporated under vacuum, the oil formed was dissolved in acetonitrile, filtered, and the solvent removed under vacuum. The white solid formed was dried under vacuum. Yield: 56%. M. P.: 156–158 °C. Anal Calcd. (%) for C_18_H_23_N_3_PAu (509.33 g mol^−1^): C, 42.5; N, 8.3; H, 4.5. Found: C, 42.2; N, 8.1; H, 4.6. MS ES (m/z) 510.13 (Au(pben)(PEt_3_))^+^. IR (KBr, cm^−1^): 3049w, ν(C-H); ν(PEt_3_), 1587 m, 1499 w, 1420 s; 1275 m, 1140 m, 1044 m, 993 m, 820 m. UV-vis (DMSO): λ_max_: 313 nm. ^1^H NMR (CDCl_3_, ppm): δ 8.52 (d, 1H, H6′), 8.47 (d, 1H, H3′), 7.65–7.85 (m, 3H, H7-H4-H4′), 7.23 (d, 1H, H5-H6), 1.31 (dt, 9H, PEt_3_), 1–93 (m, 6H, PEt_3_). ^31^P NMR (CDCl_3_, ppm) 26.91 (s). E_ox_ (at 0.02 V s^−1^) = −0.597 V; E_red_ (at 0.02 V s^−1^) = −0.811 V. Recrystallization of **2** from acetonitrile afforded crystals suitable for X-ray crystallography.

### 4.4. Cell Culture

The human neuroblastoma SH-SY5Y cell line used in this study was purchased from the American Type Culture Collection (ATCC), number CRL2266. Cells were grown in Dulbecco’s modified Eagle’s medium: Nutrient Mix F-12 (DMEM/F-12) supplemented with 10% fetal bovine serum (FBS), 1% glutaMAX, 100 U/mL penicillin and 100 μg/mL streptomycin. Cells were maintained at 37 °C in a humidified atmosphere of 5% CO_2_ and 95% air. Cells were dissociated weekly using 0.05% trypsin/EDTA when they reached a confluence of 80%. All reagents were provided by Thermo Fisher Scientific (Waltham, MA, USA).

### 4.5. Cytotoxic Effects

SH-SY5Y cells were seeded into 96-well plates at a density of 5 × 10^4^ cells per well and allowed to grow for 24 h. Cells were treated with compounds **1** and **2** at different concentrations (0.01, 0.1, 0.5, 1, 2.5, 5, 7.5, and 10 μM) for 6 and 24 h. The cytotoxic effect of compounds was evaluated with MTT (3-(4,5-dimethyl thiazol-2-yl)-2,5-diphenyl tetrazolium bromide) assay [54,55,56]. SH-SY5Y cells were rinsed with saline solution and 200 μL of 500 μg/mL MTT (Merck, Darmstadt, Germany) dissolved in saline buffer were added to each well. Following 1 h of incubation in an orbital shaker at 37 °C and 300 rpm, SH-SY5Y cells were disaggregated with 5% sodium dodecyl sulfate. The absorbance of formazan crystals was measured at 595 nm with a spectrophotometer plate reader. Saponin at 1 mg/mL was used as cell death control. The concentration of compound that produced a 50% inhibition of cell survival (half maximal inhibitory concentration, IC_50_) was determined by fitting the data with a log(inhibitor) vs response model using GraphPad Prism 6 software.

### 4.6. Flow Cytometry Analysis of Cell Death Type

The cell death type induced by compounds was determined with an Annexin V-FITC Apoptosis detection kit (Immunostep, Salamanca, Spain) following the manufacturer’s instructions [56]. Cells were seeded in 12-well plates at 1 × 10^6^ cells per well and treated with compounds at IC_50_ concentrations for 6 and 24 h. Then, cells were washed, resuspended in PBS (Phophate-Buffered Saline), and 5 μL of Annexin V-FITC and Propidium Iodide (PI) were added to each tube. Cells were incubated for 15 min in the dark and analyzed by flow cytometry using the ImageStreamMKII instrument (Amnis Corporation, LuminexCorp, Austin, TX, USA). The fluorescence of 10,000 events was analyzed with IDEAS Application 6.0 software (Amnis Corporation, LuminexCorp). The percentages of apoptotic cells, including early apoptotic cells (Annexin-FITC positive and PI negative) and late apoptotic cells (Annexin-FITC and PI-positive), and necrotic cells (Annexin-FITC-negative and PI-positive), were calculated. To further confirm if apoptotic cell death was occurring, SH-SY5Y cells were preincubated with the pan-caspase inhibitor Z-VAD-FMK (Merck) for 24 h. Then, the assay was carried out as described above. Staurosporine (STS) (Merck) was used as a positive control in all the experiments [56].

### 4.7. Caspase-3 Assay

Analysis of caspase-3 activity in SH-SY5Y cells after exposure to gold complexes was carried out using the EnzChek Caspase-3 Assay Kit (Thermo Fisher Scientific), following the manufacturer’s instructions [56]. Cells were seeded in 12-well plates at 1 × 10^6^ cells per well and treated with compounds at IC50 concentrations for 6 and 24 h. Then, cells were lysed with 50 μL lysis buffer (10 mM TRIS, pH 7.5, 0.1 M NaCl, 0.01% CHAPS (3-[(3-chloramidopropyl)dimethylammonio]-1-propanesulfonate), 1 mM EDTA and 0.01% TRITON™ X-100), resuspended and centrifuged. The pellet was resuspended in 50 μL of reaction buffer (20 mM PIPES, pH 7.4, 4 mM EDTA, 0.2% CHAPS), 10 μL of 1 M DTT (dithiothreitol), and 590 μL of H_2_O. Then, 20 μL of 0.2 mM Z-DEVD–AMC substrate (7-amino-4-methylcoumarin-derived substrate) were added, and lysates were incubated for 30 min at room temperature. Finally, the fluorescence was monitored with a spectrophotometer plate reader (excitation/emission 342/441 nm). Signals were normalized by protein concentration, which was quantified by the Bradford method. Briefly, 2 µL of each lysate were added to 200 µL of Bradford reagent, and absorbance was measured at 590 nm. Protein concentration in samples was determined using a standard curve with known concentrations of bovine serum albumin. Experiments were carried out three times in triplicate, and STS was used as a positive control.

### 4.8. Determination of ROS Production

Intracellular ROS production was evaluated with the fluorescent dye carboxy-H_2_DCFDA [57]. Cells were treated with complexes 1 and 2 at concentrations between 0.01 and 10 μM for 24 h. After this time, cells were washed with medium without serum, and 20 μM carboxy-H_2_DCFDA was added. The plate was incubated in an orbital shaker for 1 h at 37°C and 300 rpm. Then, PBS was added to each well, and cells were incubated for 30 min before measuring the fluorescence with a spectrophotometer plate reader (495 nm excitation and 527 nm emission). Experiments were performed at least three times by triplicate. The known oxidant tert-butyl hydroperoxide (TBHP) at 75 μM was used as a positive control to validate the assay.

### 4.9. Statistical Analysis

Data are presented as mean ± SEM. Statistical differences were evaluated by Student’s *t*-test with Graph Pad Prism 6 software. Statistical significance was considered at *p* < 0.05.

### 4.10. Crystallographic Studies

Crystals **1** and **2** were obtained as mentioned above. Data for **1** and **2** were collected at 293 K for **1** and 100 K for **2** on a BRUKER CCD Smart diffractometer, using graphite-monochromated Mo-Kα radiation (k = 0.71073 Å) and corrected for absorption effects by SADABS [58]. The structures were solved by the Patterson method [59], and successive Fourier synthesis gave the location of heavy atoms. All hydrogen atoms were included in the model at geometrically calculated positions and refined on F^2^. Diffuse scattering reflections due to the disordered methanol solvent molecules in **1** were corrected by SQUEEZE [39]. One of the PEt_3_ groups present in the structure of **2** also presents disorder, which was modeled considering two alternative positions (occupancy factors of 64 and 36%) for the Et groups and using common anisotropic factors for both carbon atoms. Atomic scattering factors were taken from International Tables for X-ray crystallography [60].

Molecular graphics were generated by ORTEPIII [61] and MERCURY [62]. Crystal and structure refinement data are reported in Table 4.

## 5. Conclusions

Two new gold(I) complexes containing benzimidazole and two different phosphines have been prepared and characterized. The complex that incorporates triethylphosphine ligand (**2**) is more cytotoxic against neuroblastoma SH-SY5Y cells than the complex with triphenylphosphine ligand (**1**). Our studies show that complexes **1** and **2** induce apoptosis through both caspase-dependent and caspase-independent mechanisms. In the seek of novel anticancer agents through different mechanisms than cisplatin or other Pt(II)/Pd(II) compounds, the pro-apoptotic effects of complexes **1** and **2** in neuroblastoma cells make them promising molecules for further anticancer studies.

## Data Availability

Data is contained within the article or Appendix A.

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
