# Peer review of "Combined Effect of Caspase-Dependent and Caspase-Independent Apoptosis in the Anticancer Activity of Gold Complexes with Phosphine and Benzimidazole Derivatives"

_pharmaceuticals, 2020, doi:10.3390/ph14010010_

Round 1

Reviewer 1 Report

This high-standard well-edited manuscript gives an extensive and in-depth account on gold-phosphine complexes including synthesis, structure, electrochemical properties and mechanism of anticancer activity. The research was conducted appropriately, the authors provide a detailed structural analysis and discussion of biological assays carried out on the complexes. The results are presented in clear form and supported by the experiments. I feel that only the bonding between Au(1) and N(3) - that in principle may have influence on ROS-generation, consequently on bioactivity - remains a bit of an uncertain point in structural characterisation of these complexes. Although is solid phase the sum of their van-der Waals radii is shorter than their interatomic distance, prior to final acceptance it seems to be necessary to carry out comparative 1H-15N HMBC measurements in solution on the complexes  and the free ligand "b" focusing on N(3) chemical shift that must be highly dependent on the  strength of the coordination towards the Au(I)-center. 

Author Response

REVIEWER: This high-standard well-edited manuscript gives an extensive and in-depth account on gold-phosphine complexes including synthesis, structure, electrochemical properties and mechanism of anticancer activity. The research was conducted appropriately, the authors provide a detailed structural analysis and discussion of biological assays carried out on the complexes. The results are presented in clear form and supported by the experiments. I feel that only the bonding between Au(1) and N(3) - that in principle may have influence on ROS-generation, consequently on bioactivity - remains a bit of an uncertain point in structural characterisation of these complexes. Although is solid phase the sum of their van-der Waals radii is shorter than their interatomic distance, prior to final acceptance it seems to be necessary to carry out comparative 1H-15N HMBC measurements in solution on the complexes and the free ligand "b" focusing on N(3) chemical shift that must be highly dependent on the  strength of the coordination towards the Au(I)-center. 

OUR RESPONSE: The authors are very much grateful to this reviewer for his/her valuable positive comments. We agree with the reviewer that comparative 1H-15N HMBC would shed light on the strength in solution of the contact between N3 and the gold ion. We have conducted the experiments both at room temperature (298 K) and at low temperature (278 K), but, unfortunately, we have not found 1H-15N correlation. One explanation is that the concentration of the samples in chloroform was not sufficient to detect the signals, but we were limited by the solubility of these complexes in this solvent to register 1H-15N HMBC signals.

Reviewer 2 Report

Comments for author:

  • Could you add the graphic abstract?
  • Some of the keywords are general
  • Line 23: " Cells were incubated with compounds for 24 h with concentrations ranging from 100 μM to 1 nM" but in the material and methods you mentioned " (0.01, 0.1, 0.5, 1, 2.5, 5, 7.5 and 10 μM)
  • Two benzimidazole gold(I) derivatives containing triphenylphosphine (1) ortriethylphosphine (2); could you add the chemical structures of the two compounds
  • "The pharmacological use of gold compounds for treatment of different diseases is documented since ancient times"; please support your statement with references or double check if it is true.
  • Line 45: please remove "for instance"
  • Line 53: "In one way or another, the anticancer activity of these gold compounds seems to follow different mechanisms than cisplatin or other Pt(II)/Pd(II) compounds, and they also exhibit cytotoxic activity against cells from tumor lines which are resistant to cisplatin"; please explain in more details and remove " In one way or another,"
  • Line 62: " The replacement of the S‐donor ligands by N‐donor ligands afforded a new family gold complexes that revealed interesting anticancer activities" please give examples
  • Scheme 1 needs more information about the relationship between the Auranofin compound and the three N‐donor Hpben compound.
  • A scheme of the 2 compounds synthesis should be added
  • Stability test under different conditions of the compounds is needed, and the instructions of storage.
  • Instruments characterization, i.e. NMR should be added to the material and methods section
  • In the cell culture section, the author should describe the used cell concentration in the medium.
  • Could you add references to the experimental part
  • Why the authors choose Staurosporine as positive control for all experiments
  • In the Caspase‐3 assay, authors would mention the reagents used and their concentrations.
  • Line 360, what PBS stands for?
  • In the result section of the ROS production assay, author should provide microscope images that support their results.
  • The assignment of NMR charts should be added to the manuscript
  • The conclusion part must be added.
  • The author should follow the instructions to the authors of the journal, for example, the chemical structures.
  • English editing is highly recommended.
  • The following reference could be of benefit in building up the chemical schemes.

Saeed, A.; Larik, F.A.; Lal, B.; Faisal, M.; El-Seedi, H.; Channar, P.A. Recent resurgence toward the oxidation of heteroatoms using dimethyldioxirane as an exquisite oxidant. Synthetic Communications 2017, 47, 835-52.

Author Response

Please find author's reply to the reviewer report (reviewer 2) in the attached file.

Reviewer 3 Report

The manuscript is intreste and well prepared. I reccomend to publish in Parmaceuticals.

Author Response

We are delighted that the reviewer found our manuscript to be of interest for the readers of Pharmaceuticals.

Round 2

Reviewer 2 Report

Dear Editor

Yes, it has been modified according to our suggestions.

I recommend the paper for publication.

Kindest regards, Hesham